# A Review of Non-Destructive Evaluation (NDE) Techniques for Residual Stress Profiling of Metallic Components in Aircraft Engines

Zhaoyu Shao [1], Chengcheng Zhang [1], Yankai Li [2], Hai Shen [2], Dehan Zhang [2], Xudong Yu [2,*] and Ying Zhang [1]

[1] Aero Engine Corporation of China, Commercial Aircraft Engine Co., Ltd., Shanghai 200241, China
[2] School of Astronautics, Beihang University, Beijing 100191, China
* Correspondence: yxudong@buaa.edu.cn

**Abstract:** Residual stresses are one of the main factors determining the failure of aircraft engine materials. It is not possible to reliably and accurately predict the remaining service life of aircraft engine components without properly accounting for the presence of residual stresses. The absolute level and spatial distribution of the residual stress is uncertain in aircraft engines because the residual stress profile is highly susceptible to variations in the manufacturing process. In addition, residual stresses keep evolving under complex thermal-mechanical loadings. Non-destructive techniques are desired by the aerospace industries for the regular monitoring of subsurface residual stress profile in aircraft engine components. The insufficient penetrating capability of the only currently available non-destructive residual stress assessment technique X-ray diffraction has prompted an active search for alternative non-destructive techniques. This paper provides an overview of the principle, practical applications, advantages, and limitations of four categories of nondestructive (diffraction, ultrasonic, and electromagnetic) techniques for residual stress profiling of metallic components in aircraft engines.

**Keywords:** nondestructive; residual stress; aircraft engine; diffraction; ultrasonic; eddy current; Hall coefficient





## 1. Introduction

Residual stresses are those stresses that remain in a solid material even in the absence of external loading or thermal gradients. Residual stresses form a balanced force system within an object, as all forces and moments acting on one plane through the entire object must sum to zero. Residual stresses can be categorized into 'macro-stress' and 'micro-stress' according to the length scale over which they equilibrate, and in real materials the actual residual stress comes from the superposition of these types. The 'micro-stress' extends within one grain or even only a few interatomic distances. The 'macro-stress', which extends over a large distance (from mm upwards), is what is classically considered as residual stress [1–3]. Residual stresses are ubiquitous in aircraft engine materials, and residual stress can be developed in almost all manufacturing processes and during the service life of the manufactured component. Although unavoidable and even detrimental to the material performance (tensile residual stress), residual stresses are sometimes designed on purpose for the beneficial effect on the materials (shot peening, laser shock peening, quenching) or for meeting the engineering requirements (assembly). Compressive residual stress is usually beneficial because it tends to close the micro-cracks and thus impedes the crack propagation, while tensile residual stress has a contributory effect on material fracture and is therefore harmful [4]. The main sources of residual stress of an aircraft engine include plastic deformation, thermal origins, and assembling components. Plastic deformation occurs in most production processes such as forging, rolling, bending, drawing and extrusion, and in service during surface deformation. Further, surface enhancement

techniques, including shot peening, laser shock peening, cold plasticity burnishing, deep cold rolling, and plastic hole expansion, also produce characteristic compressive residual stress profile through plastically deforming the material surface. Typical residual stress profile of surface enhanced aircraft engine materials can be seen in Bozdana (2005) and Wang (2019) [5,6]. Thermal misfit stresses arise due to temperature gradients within a body. Thermal misfit stresses can be developed in the rapid cooling (quenching) components and the in-service high temperature turbine components. A completely different category of residual stresses in a structure is due to the assembling of components to form a single structure. In many cases, bolted connections are involved. The residual stresses in the structure depend on the dimensional tolerances of the components.

Residual stress plays a key role in engineering failures such as fatigue, fracture, creep, and corrosion, and is therefore an important consideration factor in the structural integrity assessment [7–10]. Insufficient knowledge of residual stress level renders unnecessarily conservative and thus much more expensive engineering designs. Residual stress profiling of aircraft engine components is necessary in analysing the failure of key structures, validating and improving manufacturing processes/subsequent surface enhancements, and predicting remaining service life. The residual stress information can also be used as an input to understand the performance and integrity of the material and component particularly of interest to engineers [8,9]. 'Engine rotor life extension (ERLE)' and 'a concurrent approach to manufacturing induced part distortion in aerospace components (COMPACT)' are two well-known programmes in aerospace industries with a particular focus on investigating the residual stress measurement technologies and the residual stress effect on material behaviours.

The currently available residual stress measurement techniques are not perfect for the very expensive aircraft engine components. Destructive techniques such as deep hole drilling and centre hole drilling are commonly employed for residual stress measurement because of their fast speed and reliability. However, destructive measurement techniques are only applied to the components with the most critical and/or frequent occurrence of failure because of the damage to the material. Therefore, the demand for non-destructive residual stress measurement techniques is becoming more critical [11]. X-ray diffraction (XRD) is a mature non-destructive residual stress measurement technique, with its penetration depth limited to an extremely thin (less than 20 $\mu$m) surface layer for aircraft engine materials. However, the induced residual stress in aircraft engines can easily extend to more than 1 mm depth. In order to obtain the depth profile of residual stress, layer removal by electropolishing or etching is required, which also makes it destructive to the material. It is not economical to measure residual stress profile by the aforementioned techniques due to the destruction to the material, which is unacceptable by the aerospace industries for the required routine inspections. Therefore, there is growing interest in developing alternative cost-effective and portable non-destructive techniques for both industries and academic communities. A residual stress profile instead of a volume average value provides more explicit and valuable information for aero-engine components. The desired depth resolution of 10–100 um is required for the construction of residual stress profile. The measurement accuracy of the residual stress measurement techniques should be better than 10% of the material yield strength, which is the residual stress measurement uncertainty. First of all, benchmarking of existing non-destructive methods that comprise diffraction techniques, ultrasonic techniques, magnetic techniques, electromagnetic methods, and thermal techniques is performed in order to identify the promising techniques. The magnetic techniques such as Barkhausen noise and residual magnetic field are applicable only to ferromagnetic materials, which are barely used in modern aircraft engines. At present, the thermal techniques based on thermoelastic or thermoelectric effects are only applicable for surface residual stress measurement rather than depth profiling. From the benchmarking of existing non-destructive techniques, the selection of suitable techniques for non-destructive residual stress profiling is based on criteria including penetration depth and resolution of the techniques, material limitations, and portability. Therefore, this paper reviews the

selected non-destructive techniques, including diffraction, ultrasound, eddy current, and Hall coefficient, with particular focus on their depth profiling capability.

## 2. Diffraction Techniques

The principle behind the diffraction techniques for residual stress measurement is based upon Bragg's law, from which residual stresses are deduced through analysing the diffraction pattern with appropriate algorithms. The atoms in crystalline materials are positioned at regular intervals. As shown in Figure 1, the path difference between two incident beams 1 and 2 reflected at two adjacent atomic planes is $2d \sin \theta$. Constructive interference occurs when the path difference is equal to integer multiple of the X-ray wavelength $n\lambda$. The first order diffraction of a given reflecting plane is of greater importance because of its higher intensity compared to corresponding higher orders, and is usually targeted for analysis [12]. Therefore, the Bragg diffraction condition is usually written as

$$2d \sin \theta = \lambda, \tag{1}$$

where $d$, $\theta$, and $\lambda$ represent the lattice spacing of the material under test, the diffraction angle, and the X-rays' wavelength, respectively.

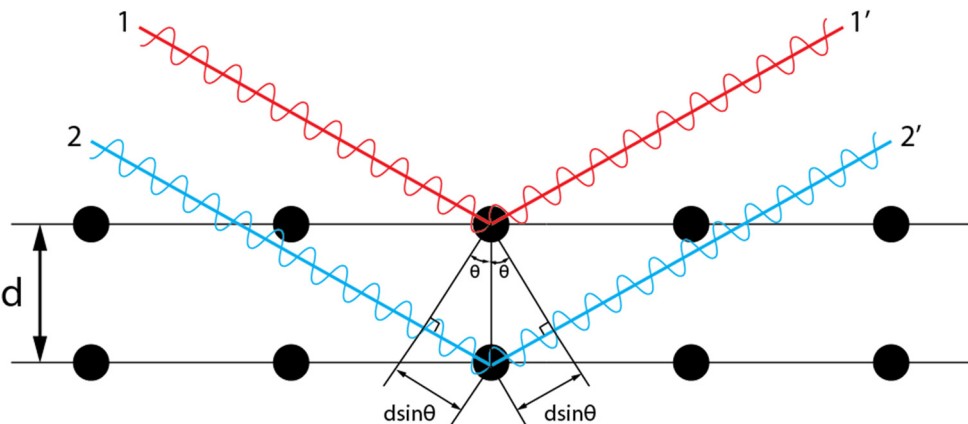

**Figure 1.** Diffraction of X-rays on two atomic planes.

The presence of residual stress will push the atoms to new equilibrium positions and thus cause the atomic plane distance change, which requires the diffraction angle changed to satisfy the constructive interference condition. When single-wavelength incident waves are used, the tensile stress will cause the diffraction peaks to shift at lower angles of incidence, while compressive stress does the opposite. By differentiating Equation (1), the relationship between the elastic strain ($\varepsilon$), lattice spacing ($d$), and diffraction angle ($\theta$) can be calculated by Equation (2):

$$\varepsilon = \frac{\Delta d}{d} \approx -\frac{\Delta \theta}{\theta}. \tag{2}$$

### 2.1. X-ray Diffraction

One of the most widely used non-destructive methods for residual stress measurement has been X-Ray Diffraction (XRD), which can be applied to any crystalline material. The conventional X-ray is generated when the electrons are released by a hot cathode collide with an anode metal target (common ones are tungsten and copper) [13,14]. There is a wide range of commercially available instruments for residual stress measurement. With the instrumental advancement, even portable devices are capable of measuring residual stresses out in the field or on large structures. As a reliable technique, X-ray diffraction is usually used to validate the numerical simulation of residual stress distributions in material processes or to compare with the results obtained by other techniques. The X-ray diffraction technique utilizes the interatomic spacing of lattice planes near the surface as the gauge

length for strain measurement [15]. The presence of residual stress will push the atoms to new equilibrium positions and thus cause the atomic plane distance change, which requires the diffraction angle changed to satisfy the constructive interference condition. As shown in Figure 2, when single wavelength incident waves are used, the tensile stress will cause the diffraction peaks to shift at lower angles of incidence, while compressive stress does the opposite.

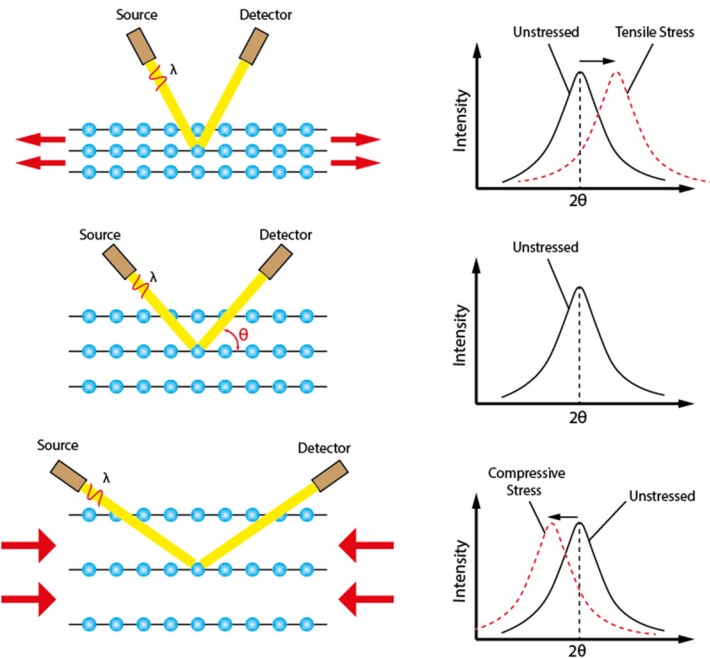

**Figure 2.** X-ray intensity peak shift with tensile and compressive stress.

XRD measurement of residual stress has become a standard procedure in laboratories and industries to measure the stress distribution of surface-treated aircraft engine components by shot peening, laser shock peening, cold rolling, ultrasonic shot peening, etc. [16–19]. As the material is subjected to a cold working process, there is a reduction in the size of the perfectly crystalline regions or crystallites, while there is an increase in the average micro-grain in the crystal lattice that produces broadening of diffraction peaks. The broadening of diffraction peaks can be conveniently measured as a means of quantifying the level of cold work that allows the separation of residual stress and cold work in surface treated components [20]. X-ray diffraction has proved to be one of the most reliable techniques for non-destructively measuring residual stresses in these aircraft engine materials.

The conventional X-rays can penetrate only up to a few tens of microns into the crystalline materials because of the exponential decay in intensity. Therefore, layer removal of material by either electro-polishing or etching is required in order to obtain the residual stress distribution over a larger depth. However, the stepwise surface layer removal process not only destroys the specimen under investigation, but it is also time-consuming. Common factors that might cause measurement error during residual stress measurement are stress constant selection, diffractometer focusing geometry, diffracted peak location, cold working, texture, grain size, microstructure, and surface condition [21]. The crystallographic texture will cause peak intensity variations of the measurement system and, if coupled with material anisotropy, will lead to oscillations in the $d$ vs. $\sin^2 \psi$ plot. The texture is a crucial consideration, and several methods have been proposed by researchers [22–25]. Another difficulty of XRD measurement of residual stress relates to the need for high diffraction angles measurement accuracy, which requires precise sample/instrument alignment and diffraction peak location. Fine-grained materials are preferred when implementing XRD

residual stress measurement, as they can produce diffraction peaks with suitable intensity without the influence of the back reflection from the sample surface [26,27].

### 2.2. Synchrotron X-ray Diffraction

Synchrotron X-ray diffraction is a promising non-destructive technique for analysing tri-axial residual stress state and material texture with high penetration depth and excellent resolution [28,29]. This method enables access to certain areas which are not accessible by traditional X-ray diffraction and can reliably achieve the precision of strain measurement up to $10^{-4}$.

Synchrotron X-rays are generated in a specialist synchrotron ring which consists of linear accelerator, the booster, and the storage ring, as shown in Figure 3. The electrons start their journey from the linear accelerator and then transfer to the booster to gain energy until their speed is close to the speed of light, then they are injected into the storage ring. When the electrons are circulating around the storage ring, they pass through specially designed magnets called insertion devices that cause the electrons to wiggle or undulate, producing intense X-ray beams. The X-ray beams will be channelled into the experimental stations, and each has an optical hutch, an experimental hutch, and a control cabin [30–33].

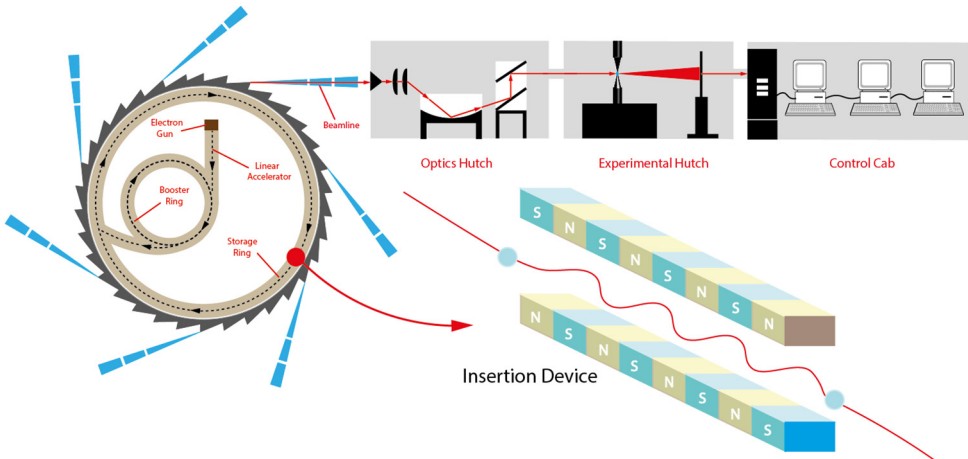

**Figure 3.** Schematic of synchrotron facility.

The synchrotron X-rays illuminate the specimen with two configurations: the reflection measurement and the transmission measurement, as shown in Figure 4. The reflection geometry (a) is suitable for near-surface normal stress measurement, as the attenuation along the path limits the maximum penetration depth. The transmission configuration (b) is used for in-plane stress measurement when the gauge volume length is similar to the sample thickness.

The synchrotron beamlines can be one million times more intense than conventional X-rays. Therefore, synchrotron diffraction has excellent penetration depth up to tens of millimetres and can provide a 3D map of strain distribution in engineering components. The scattering angles $2\theta$ generated by high energy beams typically range from $4°$ to $10°$ and therefore result in elongated, diamond-shaped gauge volumes, as shown in Figure 4. A stress-free sample needs to be extracted from the component at the measurement location in order to obtain stress-free lattice spacing $d_0$. The presence of residual stress will distort the Debye–Scherrer rings result from diffraction from numerous crystallites within the gage volume captured by 2D detectors and the corresponding strain can be obtained by analysing the change of ring radius [34,35]. The minimum size of the gauge volume is usually controlled to guarantee a sufficient number of grains scattering within the gage volume and thus obtain uniform distributed diffraction rings.

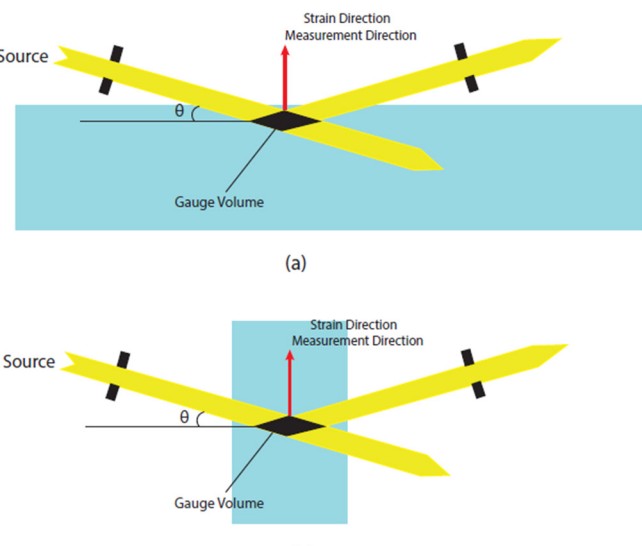

**Figure 4.** (**a**) Reflection measurement configuration. (**b**) Transmission measurement configuration.

Two types of synchrotron X-rays are used for residual stress measurement based on their different ways of exploiting Bragg's law: the monochromatic beam, which contains only single wavelength X-rays, or the white beam, which contains full spectrum photons. When monochromatic beams are used, the detector needs to scan the $2\theta$ region to obtain the peak intensity position. Therefore, this method is also called the angular dispersive method. The photons with various wavelengths in the white beam have different energies according to the Planck–Einstein relation:

$$\lambda = \frac{hc}{E}, \tag{3}$$

where $h$ is the Planck constant, $c$ is the speed of light in vacuum, and $E$ is the energy of the photon [36]. For this type of measurement, the white beams illuminate the specimen at a fixed angle and an energy sensitive detector must be used to collect the diffraction pattern. Therefore, this method is also termed energy dispersive method [2,37–42].

The last two decades have witnessed the widespread application of synchrotron diffraction to residual stress measurement in aircraft engine materials [43–46]. The monochromatic method has been employed by several researchers for assessing the residual strains as well as texture due to the high resolution of the Debye–Scherrer rings [42,47–52]. The alternative energy dispersive method, which enjoys much higher popularity, can provide white beams with higher intensity and allows very fast data collection time. The diffraction patterns obtained by this method reflect information about both residual stress and the texture at the same time [53–57]. Energy dispersive synchrotron diffraction is a versatile and powerful tool for the analysis of the tri-axial residual stress state, especially in bulk materials [58–61]. The measurement of residual stress by synchrotron diffraction requires a specialist facility with very long lead time and high cost. It may not be possible to measure the strain in certain directions because of the elongated gauge volume inside the sample, thus losing the stress information in these directions. Since synchrotron X-rays essentially interact with the electron cloud surrounding nuclei due to their electromagnetic nature, the applicability of this method is limited when applied to some metals because the attenuation increases with the atomic number [62,63].

### 2.3. Neutron Diffraction

Neutron diffraction is also a useful member of the diffraction family for studying the residual stresses in bulk materials due to its high penetration capability, and has become a well-established technique during the past few decades of development [62,64]. The

residual stress measurements are performed in either the conventional monochromatic angular scanning or white beam time-of-flight (TOF) configuration, which mainly depends on the type of neutron source [65]. The monochromatic angular scanning is usually found at reactor neutron source, and it uses neutrons of fixed wavelength, as shown in Figure 5. The monochromatic neutron beam is obtained by impinging the incident neutron on a monochromator. The diffracted neutron beam from the specimen is obtained by a detector at a fixed angle where Bragg condition is satisfied, and the strains can be obtained by evaluating the peak shift just as the XRD technique. Time-of-flight measurement is usually used at spallation neutron facilities which produce neutrons with various wavelengths, namely the white beams. The detectors can be oriented at a constant angle, usually desirable at 90 degrees to the incident beam to ensure a cuboid gauge volume. Unlike the synchrotron x-rays which reach a speed close to the speed of light, the speed of a moderated neutron is comparable to the speed of sound in the air, and the time of flight can be easily obtained. According to De Broglie's relation:

$$p = m_n v_n = \frac{h}{\lambda},\tag{4}$$

and Bragg's law, as expressed in Equation (1). The time of the neutron travelling down the flight path is expressed by:

$$T = \frac{L_n}{v_n} = d\left(\frac{2m_n L_n \sin\theta}{h}\right).\tag{5}$$

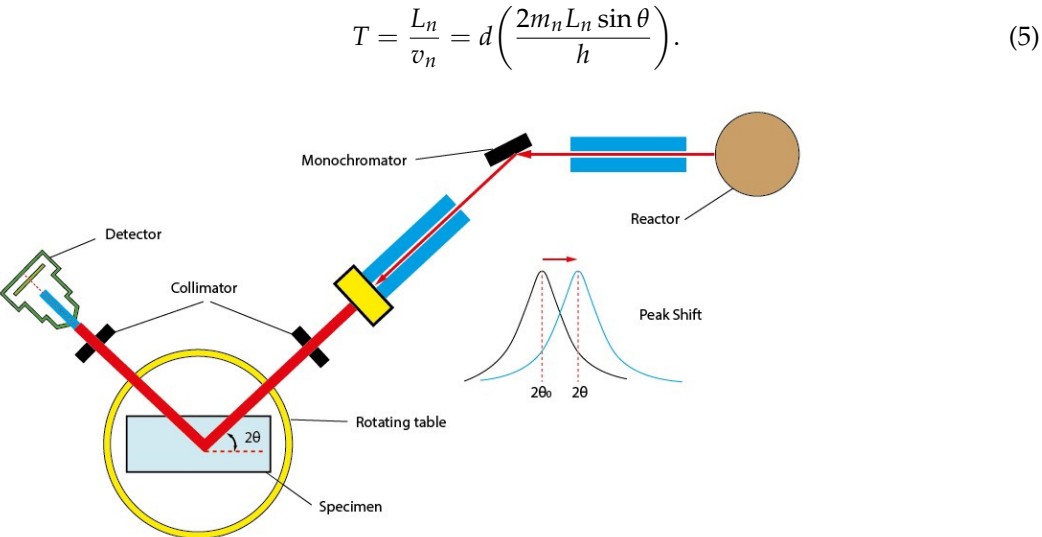

**Figure 5.** Schematic of monochromatic 2θ scanning method.

The neutron diffraction technique has already been used for in situ and ex situ residual stress measurement of aircraft engine components, including blades [66] and compressor/turbine discs [67–71], etc., due to its ideal penetration depth. Residual stress measurements by neutron diffraction are frequently applied to aero-engine materials like IN718 [72–75], IN615 [76], DD10 [77–79], Ti6Al4V [46,80–83], etc. Considering that neutron diffraction is an expensive technique, it is usually desirable to implement this technique of critical structures with complex residual stress status. The neutron diffraction technique for residual stress measurement requires intense neutron beams, available only at medium or high-flux reactor or at an accelerator-based, usually time-pulsed, with a neutron source, which strictly limits the portability of this technique and also requires a long lead time. The spatial resolution of neutron diffraction is poorer due to its larger gauge volume comparing with the synchrotron diffraction and the conventional X-ray diffraction methods, and the spatial resolution is compromised by the measurement time [62]. Neutron diffraction technology is mainly specialized in residual stress measurement at higher material depths. It is challenging to obtain residual stress profile information below 100 μm depth due to its limited spatial resolution. Conventional X ray diffraction is capable of measuring

the nearest 20μm residual stress below the surface. Synchrotron X ray diffraction can fill the important near-surface gap between what is possible with neutrons and what is accessible with traditional X-ray technique. The combination of the diffraction techniques enables the construction of the whole through-depth residual stress profile for aircraft engine components.

The strains can be obtained by analysing the lattice spacing spectrum converted from time-of-flight (TOF) spectrum, as shown in Figure 6. By using two opposing detectors, it is capable of simultaneously measuring two orthogonal strain components [2].

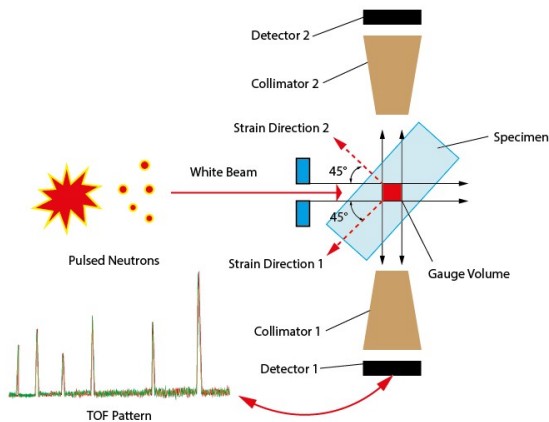

**Figure 6.** Schematic of time-of flight method.

## 3. Ultrasonic Techniques

Ultrasonic measurement of residual stress is based on the acoustoelastic effect, which describes the alteration of acoustic waves' speed traveling in solids in the presence of stress. Within the elastic limit, the speed of ultrasonic waves, which is reflected as the time of flight, exhibits a linear relationship with stress [84–86]. The complete tensor expression of the acoustoelastic effect has been proposed to link the present strain components, with corresponding velocity shifts through the material-dependent acoustoelastic constants [84,87,88]:

$$\rho v_{ii}^2 = \lambda_0 + 2\mu_0 + (2l_0 + \lambda_0)(\varepsilon_i + \varepsilon_j + \varepsilon_k) + (4m_0 + 4\lambda_0 + 10\mu_0)\varepsilon_{ij}, \tag{6}$$

$$\rho v_{ij}^2 = \mu_0 + (\lambda_0 + m_0)(\varepsilon_i + \varepsilon_j + \varepsilon_k) + 4\mu_0\varepsilon_i + 2\mu_0\varepsilon_j - \frac{1}{2}n_0\varepsilon_k, \tag{7}$$

where $\rho$ denotes the density, $\lambda_0$ and $\mu_0$ are the two Lamé constants, and $m_0$, $l_0$, and $n_0$ are the Murnaghan constants. The indices $i$, $j$, $k$ ($i$, $j$, $k$ = 1,2,3) represent the wave propagation directions, the polarization directions, and the principal strain directions, respectively ($i$, $j$, $k$ = 1,2,3). Longitudinal or shear waves will be used with pulse-echo, transmission, or pitch-catch experimental configurations. The sensitivity will depend on the propagation and particle polarization of the ultrasonic wave relative to the stress direction [1,84,89]. Calibration is necessary for different types of materials due to their varying microstructure and texture. Tensile tests are usually carried out for the calibration of the acoustoelastic effect [84]. Ultrasonic residual stress measurements are normally performed with three types of ultrasonic waves: the critically refracted longitudinal wave, shear wave in birefringence mode, and the ultrasonic Rayleigh wave [90]. It should be noted that residual stress evaluation by ultrasound is a complex issue because the velocity and propagation of ultrasonic waves will be affected by different material properties and other factors (e.g., temperature, liquid loading, coupling condition).

### 3.1. Critically Refracted Longitudinal ($L_{CR}$) Wave

The critically refracted longitudinal wave $L_{CR}$, sometimes also called surface skimming longitudinal waves, can be excited at the first critical angle just beneath the surface, and

propagates parallel to the surface at the speed of the bulk longitudinal wave. A typical setup of the $L_{CR}$ excitation system shown in Figure 7 consists of one transmitter and two receivers cased in a PMMA wedge oriented at the first critical angle, which is calculated by Snell's law when the longitudinal wave travels from the wedge to the material. Apart from the $L_{CR}$ wave skimming along the surface, a lower speed shear wave is also generated and reflects between the two surfaces of the sample [1,91]. The relationship of uniaxial stress and travel time change of $L_{CR}$ wave was derived by Egle et al. (1976) [84] as:

$$\Delta\sigma = \frac{E}{Lt_0}(t - t_0 - \Delta t_T),$$ (8)

where $\Delta\sigma$ is the stress variation, $E$ being Young's modulus, $L$ being the corresponding acoustoelastic constant, $t_0$ being travel time at the stress-free state, and $\Delta t_T$ is the travel time change due to the temperature change. The unique feature of $L_{CR}$ is its highest sensitivity to stress compared with other types of ultrasonic waves due to the maximum magnitude of acoustoelastic constant $L$ in Equation (5) that couples the corresponding velocity change and the stress parallel to the wave propagating direction. Similar to the Rayleigh wave, the effective penetration depth of the $L_{CR}$ wave (more than one wavelength) exhibits the frequency dependence, which enables the residual stress assessment at various depths [91–94].

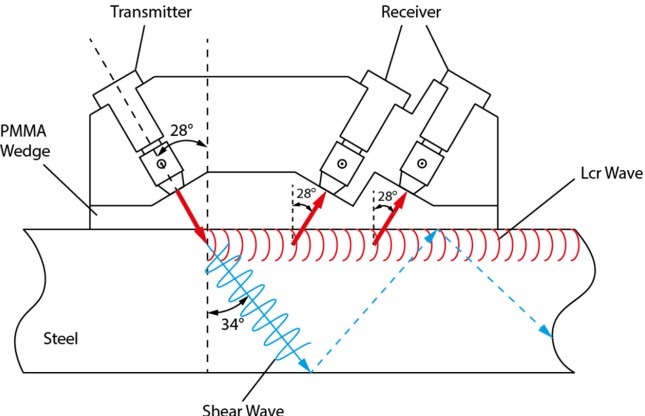

**Figure 7.** Typical set-up for generating wave.

Experimental evidence [94,95] suggested that the penetration depth of the $L_{CR}$ wave is frequency-dependent, which is promising in residual stress profiling. However, there is no theoretical work defining the penetration depth and frequency relationship of the $L_{CR}$ wave and therefore this needs to be determined experimentally. Bray et al. (1991, 2001) [94,96] studied the actual wave energy distribution through depth experimentally. They suggested the effective penetration depth to be the distance from the material surface to the peak of the penetration energy distribution curve. Javadi et al. (2013) [97,98] proposed a more straightforward way of defining the penetration depth. A groove is cut and incrementally increased through milling to produce an obstacle between the exciting and receiving transducers for the $L_{CR}$ wave transmission. When the groove reached the depth where no $L_{CR}$ signal could be detected by the receiver, the depth is considered as the penetration depth. It should be noted that propagation distance of incident $L_{CR}$ waves has to be sufficient in order to resolve from the simultaneously travelling shear wave and Rayleigh wave with similar velocities; otherwise, the penetration depth of the pure $L_{CR}$ wave would be difficult to determine experimentally.

With the proper relationship between penetration depth and frequency of $L_{CR}$ wave, the residual stress gradient model [99] can thus be proposed, as shown in Figure 8. The model treats ultrasonic coverage as a rectangular layered area. The stress variation calculated in Equation (5) with pitch-catch configuration reflects the averaged state of residual

stress ranging from specimen surface to the maximum penetration depth of $L_{CR}$ wave. Thus, the inversion of averaged stress in each layer can be obtained as:

$$\sigma_{ij} = \frac{\sigma_j D_j - \sigma_i D_i}{D_j D_i},$$

(9)

where $\sigma_i$, $\sigma_j$ and $D_i$, $D_j$ denote the varied stress penetration depths of the incident $L_{CR}$ wave in two adjacent layers, respectively. These stresses are measured at different frequencies, according to the determined frequency-dependent penetration depth.

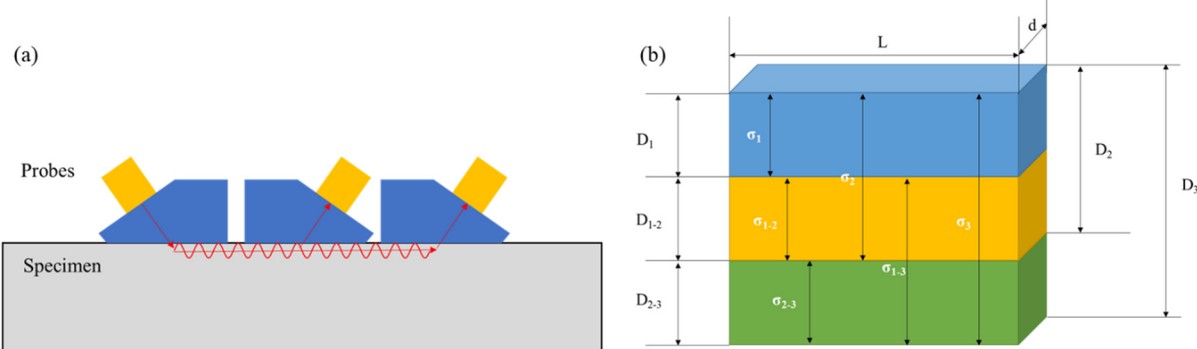

**Figure 8.** Schematics for (**a**) pitch-catch experimental configuration and (**b**) residual stress gradient model.

The $L_{CR}$ wave was extensively applied for residual stress measurement in bulk materials because it is an easy-to-use, fast, and inexpensive non-destructive technique that can perform in situ measurements with satisfactory resolution and sensitivity. The penetration and propagation characteristics make the $L_{CR}$ technique particularly suitable for measuring surface/subsurface residual stresses profiles. The calibration specimens for extracting acoustoelastic constant should have the same microstructure as the material to be tested. The microstructure influenced the accuracy of this technique, particularly in specimens that have dramatic microstructure variations such as the welded parts.

In summary, quantitative estimation of residual stress using $L_{CR}$ wave remains demanding. The accuracy of stress inversion can be immediately affected by various factors, such as the empirical frequency–penetration depth [100], and the angular deflection of incident/received ultrasonic beams [99]. Meanwhile, the measuring accuracy of velocities of $L_{CR}$ wave will also be influenced by operating temperature, specimen grain size, material texture, and surface treatments (especially cold work) [101]. Furthermore, the incident wavefield is often a combination of shear wave, longitudinal wave, Rayleigh wave, and head wave [100]. They can only be resolved upon sufficiently long-distance propagation. Nevertheless, residual stress evaluation via $L_{CR}$ wave can always be adopted as a rapid qualitative NDE method.

*3.2. Rayleigh Wave*

Rayleigh wave is a type of wave that propagates along the surface of solids without radiation loss in contrast to the $L_{CR}$ wave and shear wave with the energy concentrated within one wavelength layer beneath the surface. The acoustoelastic relationship of Rayleigh wave propagating in the principal stress directions can be expressed by:

$$\frac{\Delta V_1}{V_1^0} = K_{12}^1 \sigma_{11} + K_{12}^2 \sigma_{22},$$

(10)

$$\frac{\Delta V_2}{V_2^0} = K_{21}^1 \sigma_{11} + K_{21}^2 \sigma_{22},$$

(11)

where $K_{12}^1$, $K_{12}^2$, $K_{21}^1$, and $K_{21}^2$ are acoustoelastic coefficients, with the superscripts denoting the loading directions. The acoustoelastic coefficients can be obtained through uniaxial loading of the stress-free tensile samples, and the extracted specimens should be oriented along with the principal stress directions. The distance between the two wedges typically used in the experiments should be kept constant, even the specimen will be lengthened during the tensile test [102]. Considering the extensive efforts for the experimental calibration of acoustoelastic coefficients, researchers [103–106] proposed the theoretical calculation of acoustoelastic coefficients of Rayleigh wave from the intrinsic second and third-order elastic constants, which can also be obtained by longitudinal and shear wave velocity measurements taken on the same material. It was proved that Rayleigh waves are sensitive to stress components parallel or normal to their propagating directions, which makes it a very promising technique for measuring surface residual stress that is usually induced deliberately by surface treatment like shot peening, laser shock peening, rolling, and plastic burnishing [107].

Although the velocity change of the Rayleigh wave caused by surface treatment is less than 1%, with the instrumentation development, especially the utilization of laser measurement, the acoustoelastic effect can be measured with very high accuracy [108–110]. For example, as indicated by recent advances in the Spatially Resolved Acoustic Spectroscopy (SRAS) technique, the velocity of travelling Rayleigh wave can be measured rapidly, with a relative error as low as 0.03% [111,112]. As schematically illustrated in Figure 9a, the SRAS technique utilizes the short laser pulses (containing a broad range of frequencies) to generate surface acoustic waves (SAW) via pulsed grating pattern. The frequencies of the excited SAWs are determined by the grating space, which can be captured at the vicinity of the excitation region by another detector laser. The acoustic velocity of SAWs, i.e., Rayleigh waves, can be obtained using $v_{SAW} = f_{SAW} \cdot \lambda_{gs}$, with $f_{SAW}$ being the frequencies generated and $\lambda_{gs}$ being the prescribed grating space. A velocity map can then be built upon the test sample surface, then raster scanned, and such measurement is then repeated for each spatial point. Owing to the velocity mapping with high accuracy, the SRAS technique has been well applied to non-destructively evaluating the material microstructure and grain orientation, and the immediate comparison between the SRAS reconstruction (Figure 9b) and the EBSD image (Figure 9c) of a welded specimen show good agreement. It can be predicted that the SRAS technique would be very attractive for reconstructing the residual stress profiles, upon the influences of microstructure on the Rayleigh wave velocity can be isolated.

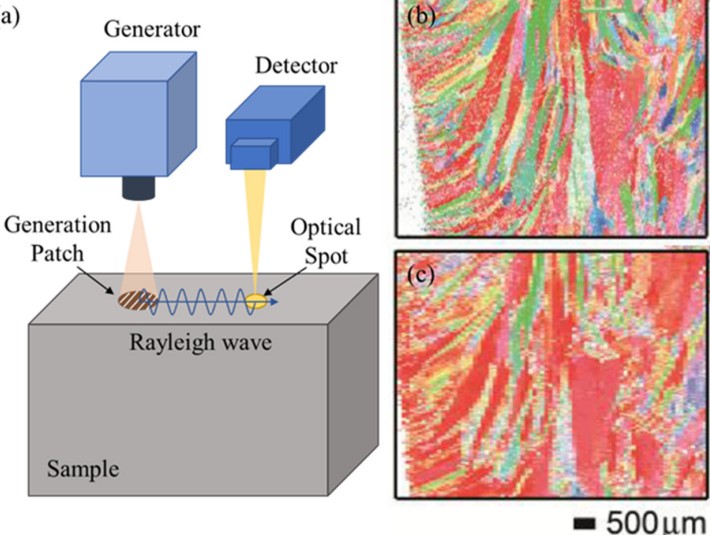

**Figure 9.** (**a**) Schematic showing the optical and acoustic working principles for the SRAS technique, (**b**) the SRAS imaging of the grain distribution of a weld specimen, (**c**) the EBSD reconstruction for the weld specimen [113].

On the other hand, it was observed by Junge et al. (2006) [114] that the Rayleigh wave polarization, which is defined as the ratio of the maximum in-plane to the maximum out-of-plane displacement of a particle on the free surface, can be an indicator of stress state, as its relative change also exhibits linear dependence on the applied stress however much more sensitive than the relative velocity change. This phenomenon is very promising to be exploited for residual stress evaluation, although further studies are required to validate the effectiveness of this method with the consideration of the influential factors such as anisotropic texture and surface roughness. It was also found that the second-order harmonic amplitude of the Rayleigh wave has a strong dependence on material plasticity, which is closely related to the residual stress [115,116]. There is great potential to explore this feature of Rayleigh wave for residual stress assessment. Liu et al. proposed a preliminary method for evaluating residual stress using nonlinear Rayleigh wave, and the nonlinearity parameter was deduced as:

$$\beta \propto \frac{A_2}{A_1^2 f^2},$$ (12)

where $A_1$, $A_2$ denote the amplitudes of the fundamental and second harmonic in frequency domain, and $f$ is the driving fundamental frequency. Initial measurements were taken on shot-peened samples, yet the separation of factors like plasticity and texture still requires systematic studies [117]. The Rayleigh wave velocity is independent of frequency when propagating in homogeneous material; however, with the presence of property gradient through depth such as residual stress, cold work, etc., the Rayleigh wave becomes dispersive, which means the phase velocity of the Rayleigh wave exhibits frequency dependence [118]. Due to the frequency-dependent penetration depth of the Rayleigh wave, it is possible to obtain the residual stress profile by measuring Rayleigh wave phase velocity over a frequency range with appropriate inversion of the dispersion data [90,105,119–121].

According to the studies of Rayleigh wave dispersion on shot-peened specimens, a significant reduction in the dispersion was observed after thermal relaxation, which can be attributed to the disappearance of residual stress as well as cold work (including crystallographic texture) [122,123]. Cold work may be a dominating effect on the Rayleigh wave dispersion, since the surface enhancement techniques usually introduce a large amount of cold work [124–126].

The Rayleigh wave exhibits scattering-induced attenuation, which can influence the velocity measurement. Besides, the surface roughness mainly influences the coupling condition between the transducer and the specimen surface, resulting in a significant error in the time-of-flight measurement. Therefore it is desirable to conduct Rayleigh wave measurement on specimens with a smooth surface, which can be achieved through grinding [127].

The crystallographic texture gradient through the thickness will also influence the Rayleigh wave dispersion and the acoustoelastic coefficients. However, the Rayleigh wave can be combined with the critically refracted longitudinal wave to give texture-independent measurements [128,129].

## 4. Eddy Current

Eddy current is a swirling current induced by magnetic induction when there is a change of an applied magnetic field to the conductors. As shown in Figure 10, when the coil with alternating current passing through and approaching a conductor surface, a secondary magnetic field will be induced against the primary magnetic field generated by the coil, thus exciting the closed-loop eddy currents whose density decays exponentially with depth. The standard penetration depth of eddy current is defined as the depth where the eddy current density drops to $1/e$ or 37% of the surface current density and can be calculated by:

$$\delta = \frac{1}{\sqrt{\pi f_c \mu_c \sigma_c}},$$ (13)

where $f_c$, $\mu_c$, and $\sigma_c$ represent frequency, permeability, and conductivity, respectively.

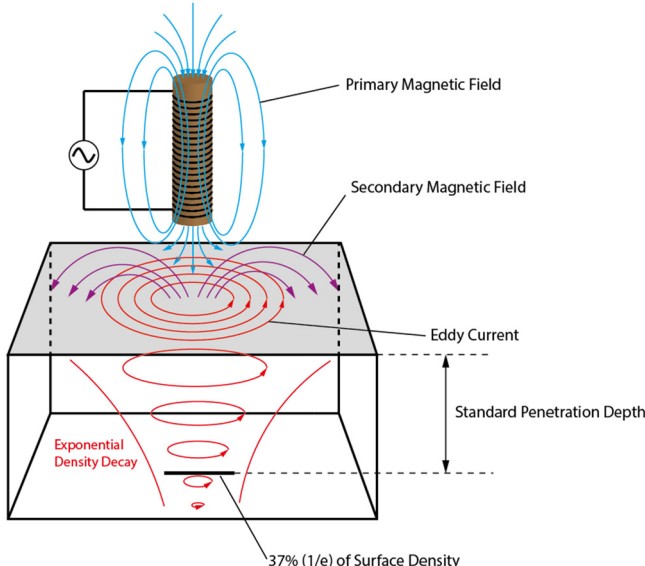

**Figure 10.** Schematic of the generation and penetration depth of eddy current.

Eddy current conductivity measurement became a promising candidate for residual stress profiling of aircraft engine materials, and a few systematic studies have been conducted since the 2000s [130–136]. This technique is based on the well-known piezo-resistive effect: there is a linear relationship between the change of resistivity/conductivity and stress the material is subjected to. Since the stress-induced apparent eddy current conductivity (AECC) change can be accurately obtained by eddy current detection and there is a frequency-dependent penetration depth of eddy current, there is great potential for reproducing the near-surface residual stress profile, especially for surface-enhanced aero-engine materials [137–139]. The development of the eddy current method is mainly aimed at obtaining subsurface residual stress distribution of specimens treated by shot peening, which induces isotropic plane residual stresses. The piezo-resistive relation for isotropic plane stress can be expressed by:

$$\frac{\Delta\sigma_c}{\tau_c} = \frac{1}{2}\left(K_{\parallel} + K_{\perp}\right), \tag{14}$$

where $\sigma_c$ and $\tau_c$ represent the conductivity and stress, respectively. $K_{\parallel}$ and $K_{\perp}$ are the parallel and normal electro-elastic coefficients, which can be accurately measured by uni-axial cyclic loading test with directional eddy current probes when the thermos-elastic is appropriately corrected [140]. For residual stress measurement on surface-enhanced components, generally, the probe is non-directional, and therefore the eddy current only measures the effective piezo-resistive effect that is the superposition of the parallel and normal piezo-resistive effects. The parallel and normal electro-elastic coefficients in most metals and alloys are opposite in sign, and therefore the net piezo-resistive effect is too weak to detect. However, it was found that the parallel and normal electro-elastic coefficients in some nickel-based superalloys, such as IN100 and Waspaloy, are both negative and in a similar large magnitude. Because of low electrical conductivity (1.5%IACS), the penetration depth of IN100 and Waspaloy is relatively high at a given frequency (about 180 μm at 10 MHz) and, therefore, the surface roughness (only a layer of a few μm) influence is limited. Furthermore, the cubic symmetric structure will not exhibit electrical conductivity anisotropy, and thus the spurious subsurface crystallographic texture does not affect the measurement. These desirable features make the eddy current technique very promising for subsurface residual stress profiling in IN100 and Waspaloy, which are widely used in aero-engine components [141]. The AECC does not change with frequency when measured

on un-peened specimens that have uniform through-thickness properties. However, the AECC increases with frequency on shot-peened specimens, and the change is greater when the surface is peened with higher Almen intensity. The increase in AECC is due to the combined effects of residual stress, cold work, and surface roughness, which are even more pronounced at a higher inspecting frequency when the eddy current is squeezed toward the surface. Since the presence of cold work and surface roughness both reduce the AECC, it was initially concluded that the residual stress is the dominating factor for the AECC increase. This conclusion was further supported by the measurements on shot-peened specimens subjected to a different level of thermal relaxation. With more thermal relaxation, the AECC gradually decreases and disappears when the residual stress is fully relaxed, although the cold work is only partly relaxed. This plausible selectivity of AECC to residual stress enables the feasibility of residual stress profiling in shot-peened nickel-based alloys by eddy current method [130,138]. Inversion procedures have been developed to convert the AECC spectrum into the depth-dependent electric conductivity profile, which can further be converted into the residual stress profile based on the piezo-resistivity effect [142–144].

The eddy current technique is also sensitive to other factors apart from residual stress, such as surface roughness, cold work, and hardness. Therefore, the major difficulty for using this technique is not its weak sensitivity to residual stress, but its selectivity.

### 4.1. Influence of Surface Roughness

At high inspection frequency, there is an apparent loss of observed conductivity as the eddy currents concentrate on the surface and follow a more tortuous route due to surface roughness. The surface roughness-induced electrical conductivity loss is dramatic for high conducting materials like copper because of their lower inspecting depth (Equation (18)) compared with nickel-based superalloys, which have only 1–2% IACS electrical conductivity. However, the surface roughness effect cannot be readily ignored in nickel-based superalloys, as the in-service components exhibit an increased surface roughness as a result of corrosion, erosion, or fretting wear. Therefore the spurious surface roughness effect needs to be corrected when conducting eddy current evaluation of in-service aero-engine components [145]. Kalyanasundaram et al. (2004) [146,147] proposed a simple analytical model approximating the rough surface as a one-dimensional sinusoidal corrugation based on the Rayleigh–Fourier method. The obtained theoretical results turned out to be in good quantitative agreement with experimental data from shot-peened copper specimens. It was suggested that annealing the specimens will remove both residual stress and cold work, and therefore the influence of surface roughness can be examined alone [148]. Johnson et al. (2005) [149] carried out statistical analyses of scanned eddy-current impedance data for IN718 specimens under various levels of shot peening and heat treatments. It is anticipated that the analysis of statistical distributions in spatial eddy-current data will help to distinguish between the effects of surface roughness and residual stress.

### 4.2. Influence of Cold Work

It was first observed in [130] that the AECC variation vanished after thermal relaxation, which removes all residual stresses, while one-third of cold work still exists. This leads to the conclusion that cold work contribution to the AECC difference can be neglected. However, attributing AECC change entirely to residual stress gives a significant overestimation on the converted residual stress profile based on the piezo-resistive effect when compared with the residual stress profile constructed by the more reliable X-ray diffraction technique. This contradiction suggests that cold work influence on the AECC change is more complex than expected. The three prior factors that cold work affects, thus resulting in overestimation, are material permeability, electro-elastic coefficient, and electric conductivity. Experiments have demonstrated that neither the material permeability nor the electro-elastic coefficient is affected significantly by cold work. However, the electric conductivity increases dramatically, probably due to unknown microstructural changes. Hillmann et al. (2009) [150] measured conductivity of a shot-peened IN718 specimen and

also a roller burnished IN718 specimen with a precision impedance analyser. It was found that the conductivity decreases with the increased peening density for shot-peened IN718 specimens, while the roller burnished samples exhibit an opposite behaviour. The main difference between these two surface treatment processes is the different level of cold work imparted to the material. Therefore, the observed phenomenon suggests that cold work has a significant effect, which is different from residual stress on eddy current conductivity. The swept-frequency eddy current measurement on shot-peened and laser shock peened IN718 samples by Lesthaeghe et al. (2013) also reflected the effect of cold work [151].

Yu et al. (2006) [152,153] proposed an explanation of cold work influence on eddy current characterisation of near-surface residual stress to address the 'contradiction' of the observed results from previous and later experiments. It was suggested that cold work cannot be considered as a single influence parameter but should be separated as type A cold work and type B cold work. Type A cold work strongly affects electrical conductivity but affects X-ray diffraction peak width much less, and decays rapidly during thermal relaxation. On the contrary, type B cold work strongly affects XRD measurements while affecting electrical conductivity much less, exhibiting weak thermal relaxation. Therefore, in the previous experiments with full relaxation samples, the AECC difference vanished because of the disappearance of type A cold work, which in fact contributed to the electrical conductivity change in the subsequent experiments. The proposed cold work effect mechanism has not been validated yet and will have to be investigated further. Hillmann et al. (2010) [154] proposed four methods of separating residual stress from cold work without considering the effect of microstructure. However, the four proposed methods have corresponding limitations. It turns out that the effects of cold work and residual stress on conductivity are not easy to separate, and further investigations need to be conducted. It seems realistic to combine two non-destructive approaches, such as eddy current with ultrasonic time-of-flight measurements.

### 4.3. Influence of Hardness and Microstructure

It was shown in [155,156] that the relationship between the electric conductivity depth profile and residual stress profile is also sensitive to the sample hardness. According to the measurements on shot-peened IN718 samples with various hardness, there is a continuous transformation of ΔAECC from positive to negative with increased hardness, while the residual stress in these samples exhibits almost no change from XRD measurement. The optical images of the samples have been examined, and it was concluded that any change of hardness and eddy current response is due mainly to $\gamma''$ precipitates or, to a lesser degree, $\gamma'$ precipitates, although other subtle changes like short-range ordering in the $\gamma$ matrix cannot be excluded. The unexpected phenomenon becomes a formidable problem for eddy current spectroscopy of residual stress in most critical aero-engine components with hardness above the level at which the ΔAECC becomes negative. Although the bulk AECC (average conductivity measured at the frequencies between 0.6 and 1.1 MHz) was suggested to correct the spurious hardness effect, clearly, there is still much to be done to understand this effect. The different hardness levels are due to the microstructure variation, which proved to have a strong effect on the eddy current signals from the shot-peened specimens. Proper compensation is necessary [157].

Chandrasekar et al. (2012) [157] conducted a systematic study of microstructural effects on eddy current responses of shot peened IN718 samples with different secondary phase contents (hardness levels). A strong dependence of the eddy current responses on the sample hardness and microstructure has been observed. The estimated conductivity deviations caused by shot peening tend to be larger than evaluated from the empirical piezoresistivity relation, even when a matched filter technique has been developed to suppress the microstructure dependency. The overestimation indicates that other mechanisms (the induced dislocations and lattice defects, formation of solute-rich atmospheres in the matrix) may contribute to the observed conductivity changes. The quantification of size and concentration of the precipitates ($\gamma'$, $\gamma''$, $\delta$ and metal carbides) and their effects on the conductivity

of nickel-based superalloys have been investigated by Chandrasekar et al. (2012) [158], Pereira et al. (2015) [159], and Nagarajan et al. (2017) [160]. Compensation of the microstructure effects on conductivity is required in order to reliably implement residual stress profiling in precipitation-hardened aero-engine materials.

## 5. Hall Coefficient

A novel galvanomagnetic NDE technique based on the Hall coefficient has been proposed as an extension of high-frequency eddy current measurements, for the purpose of residual stress assessment in precipitate hardened aerospace superalloys. This technique aims at separating the effects of cold work from that of residual stress and has been investigated by several researchers [161–163].

When an electric current is flowing through a conducting material that is placed in a magnetic field, a weak voltage will be induced across the material in a direction perpendicular to both the current and magnetic field, as shown in Figure 11. This phenomenon is known as the Hall effect. The Hall coefficient is an important material parameter that characterises the efficiency of generating the Hall electric field in the material. It can be obtained in the Hall effect experiment, as expressed by the equation:

$$R_H = \frac{V_H t}{IB},$$ (15)

in which $V_H$, $I$, $B$, and $t$ represent Hall voltage, current passing through the sample, normal magnetic flux density, and sample thickness, respectively.

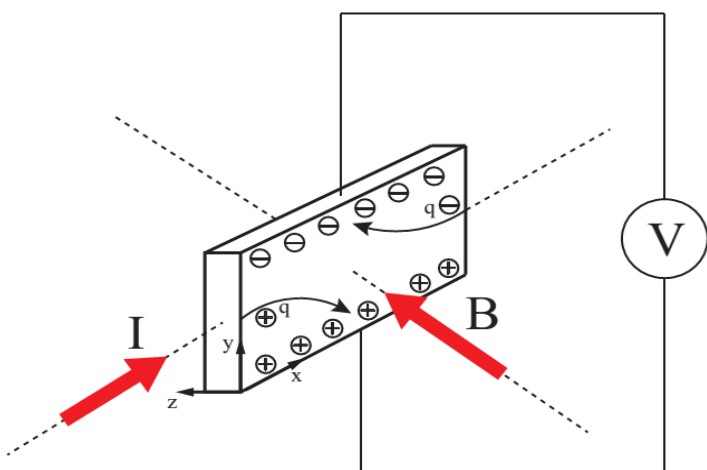

**Figure 11.** Schematic of the Hall effect.

The Hall coefficient can be calculated according to the free charge carrier approximation as [164]:

$$R_H = \frac{1}{n_c q}.$$ (16)

It is obvious that the Hall coefficient magnitude is only determined by the carrier density $n_c$, and its sign is determined by the carrier type $q$. Therefore, it is often used to characterise the carrier concentration and carrier type in the material. Conventionally, the Hall coefficient is measured by the Van der Pauw method [165,166], which requires uniform and extremely thin material so that a more pronounced Hall effect can be observed. The Hall coefficient is more difficult to measure on metals and their alloys than semiconductors, as the Hall effect is much weaker in these materials. It was first proposed by Nagy (2013) [161] that it is possible to characterise the residual stress according to the Hall coefficient variation. The Hall coefficient was obtained non-destructively through a modified four-point transfer resistance measurement. According to the initial results on aluminium specimens, the Hall coefficient changed when the stress was applied and returned

to the initial value after the stress was relieved. The result qualitatively proved the stress dependence of the Hall coefficient, although the measurement error is still big. Recently, the linear relationship between the Hall coefficient and elastic tensile stress was demonstrated on nickel-based superalloys by Shao et al. (2018) [163], Kosaka et al. (2016) [162,167], and Velicheti et al. (2017) [168], with the modified alternating current potential drop (ACPD) configuration measurement method. Therefore, it is possible to evaluate residual stress in nickel-based super-alloys or even other conducting materials based on the Hall coefficient variation with appropriate calibration. The Hall coefficient technique was proposed mainly to address the limitation faced by the eddy current technique that the cold work effect is opposing or even overshadowing the residual stress effect for precipitation hardened aircraft engine materials like IN718 [153]. However, through the Hall coefficient method, the residual stress effect and the cold work effect on the Hall coefficient are in the same direction rather than cancelling each other. It is possible to evaluate the compressive residual stress profile based on Hall coefficient measurements even in surface-enhanced aircraft engine components [169].

Although the stress dependence of the Hall coefficient has been demonstrated in several aircraft engine materials, the aforementioned demonstrative experiments for Hall coefficient measurement are still destructive, as they require cutting the specimens with limited width when measuring the Hall voltage. This is because the Hall voltage is produced only when the Hall current (normal to the conducting current in the magnetic field) is intercepted by boundaries. The boundaries can be produced by cutting the specimens with limited width, which is destructive, or confines the magnetic field area. Velicheti et al. (2018, 2017) [170,171] proposed two different approaches for non-destructive Hall coefficient measurement in plates based on a modified ACPD technique using a four-point square-electrode probe with external magnetic modulation. Both techniques were investigated analytically to validate the underlying physics and followed by numerical simulations for quantitative predictions. At low frequencies, the destructive specimen cutting can be replaced by constraining the bias magnetic field with less than one order of magnitude loss in sensitivity. At sufficiently high inspection frequencies, the magnetic field of the Hall current induces a strong enough Hall electric field that produces measurable potential differences between points lying on the path followed by the Hall current. Above a certain cut-off frequency, the sensitivity increases proportionally to the square root of frequency. The high frequency method can potentially be exploited for sub-surface residual stress profiling in surface-enhanced aero-engine components.

It was found that the proposed two methods for non-destructive Hall coefficient measurements conducted with contact sensing electrodes above 5 kHz exhibited reduced accuracy due to spurious inductive coupling between the injecting and sensing loops and the inevitably decreasing common mode rejection ratio (CMRR) of high-frequency differential preamplifiers. Velicheti et al. (2018) [172] demonstrated the feasibility of an alternative approach to broadband Hall coefficient measurement, based on inductive sensing of the circular Hall current (Corbino current) [173]. Theoretical analysis and finite element simulation were conducted for this approach, followed by experimental validation. It was found that Hall voltage induced in the sensing coils increases proportionally to the square root of frequency when the penetration of both the injected primary current and the secondary Corbino current is limited by the electromagnetic skin depth. The Hall voltage produced by a given injection current can be easily increased by an order of magnitude or more using a pair of multiple-turn sensing coils connected in series. The inductive sensing approach reduces the required CMRR of the receiving electronics, which is particularly difficult above 1 MHz when contact electrodes are used in ACPD measurements [172]. Later, the feasibility of separating the competing effects of near-surface residual stress and cold work in shot-peened fully hardened IN718 alloy in high-frequency dual-mode Hall impedance and eddy current conductivity measurements was investigated. An inversion procedure was developed accordingly to assess the sought depth profiles. It was found that the competing residual stress and cold work contributions of shot peening

can be separated using dual-mode high-frequency inspection [174]. Recently, the proposed inversion procedure has been implemented on experimental data measured on IN718 coupons shot-peened at Almen 4 A – 12 A intensities [175]. The elastic and plastic gauge factors were obtained by independent calibration tests conducted under uniaxial stress. Comparison between the dual-mode NDE results and residual stress and cold work depth profiles obtained by destructive XRD testing showed reasonable qualitative agreement, although residual stress levels were significantly underestimated, while the cold work levels were slightly overestimated. An empirical correction was applied which reduced the elastic gauge factors by 32% and increased the plastic gauge factors by 32%. This empirical correction using a single correction factor for all three peening intensities significantly improved the accuracy of the inverted NDE residual stress and cold work depth profiles. However, the dual-mode mode measurements conducted on these thermally exposed shot-peened IN718 coupons revealed that Hall coefficient and conductivity depth profiles were drastically different from numerical predictions based on XRD data and linearized gauge factors. These tests indicated that thermal exposure produces a depth-dependent change in physical properties with a penetration depth that is of the same order as that of the cold work. The measured data suggested that thermal exposure led to thermally activated microstructural changes in the highly cold worked near-surface regions of the shot-peened IN718 coupons.

According to the studies so far, the Hall coefficient technique exhibits both great promises and difficulties. The measurement of the very weak Hall voltage (nano-volt level in metals and alloys) requires a very precise system and a very stable environment, as any external disturbance may influence the measurement results. The measurement temperature should also be strictly controlled, as the temperature will influence the Hall coefficient [167,176]. The dual-mode galvanomagnetic method provides a potential solution for near-surface residual stress profiling of aero-engine materials. The adaptation of the dual-mode galvanomagnetic method for residual stress and cold work depth profiling on real aero-engine components still requires further investigation on the disturbing factors, complex geometries challenges, inspection speed, measurement automation, etc.

## 6. Future Trends of Residual Stress Profiling

Revisiting the four categories of the above-mentioned NDE techniques, it can be found that these techniques will be suited for residual stress profiling under varied inspection routines. Owing to the advancement in artificial intelligence as well as monitoring philosophy, we deem that the possible future technologies for residual stress characterisation can broadly be categorized into three strands, which include data fusion of different NDE techniques, model-based smart NDE, and potential monitoring strategy of stress states. Related techniques will aim at improving the scope and accuracy, as well as the reliability of residual stress profiling.

### 6.1. Stress Measurement via Data Fusion of Different NDE Techniques

Residual stress changes the structural responses to different NDE modalities, making it capable of characterising the stress quantities, whereas relying on only single NDE technique would compromise the scope and accuracy of the stress measurement. Therefore, it will be attractive to develop robust fusion method to shed light on the comprehensiveness of stress measurement. Methods have been studied to make decisions based on different inputs, and they are collectively called "data fusion" methods [177].

Joint Directors of Laboratories [178] define data fusion as a "multi-level, multifaceted process handling the automatic detection, association, correlation, estimation, and combination of data and information from several sources". Data fusion has been used in a wide variety of technologies, for instance, in the field of NDE. Such a fusion technique, combining the advantages of different NDE techniques, can simplify the interpretation of experimental data and associated feature extraction. Data fusion literature defines different levels at which fusion may take place: data-level, feature-level, and decision-level [179].

The first combines the original amplitudes, the second extracts compatible features and combines them, and the last and the highest level involves combining decisions derived from independent analysis of different data channels [180].

The process of data fusion for stress measurement is depicted in Figure 12. Each NDE technique herein generates signals from specified multiple sensing, and the signals corresponding to each technique need to undergo data-level fusion first to produce independent data encapsulating residual stress information. Next, the data from different techniques are registered, which means transforming the data obtained from multiple sensors into a common coordinate system [181]. Specifically, on the basis of different physical principles of NDE techniques, the data obtained by each method reflect a varied state of residual stresses. For example, the data obtained by diffraction techniques correspond to the stresses at a specific point; the data obtained by ultrasonic methods correspond to the averaged stresses in the sonic area; and the data derived by eddy current techniques would correspond to stresses at varied penetration depths. In this regard, data registration step would be beneficial for fusing stress-related information at different spatial areas and over various depths.

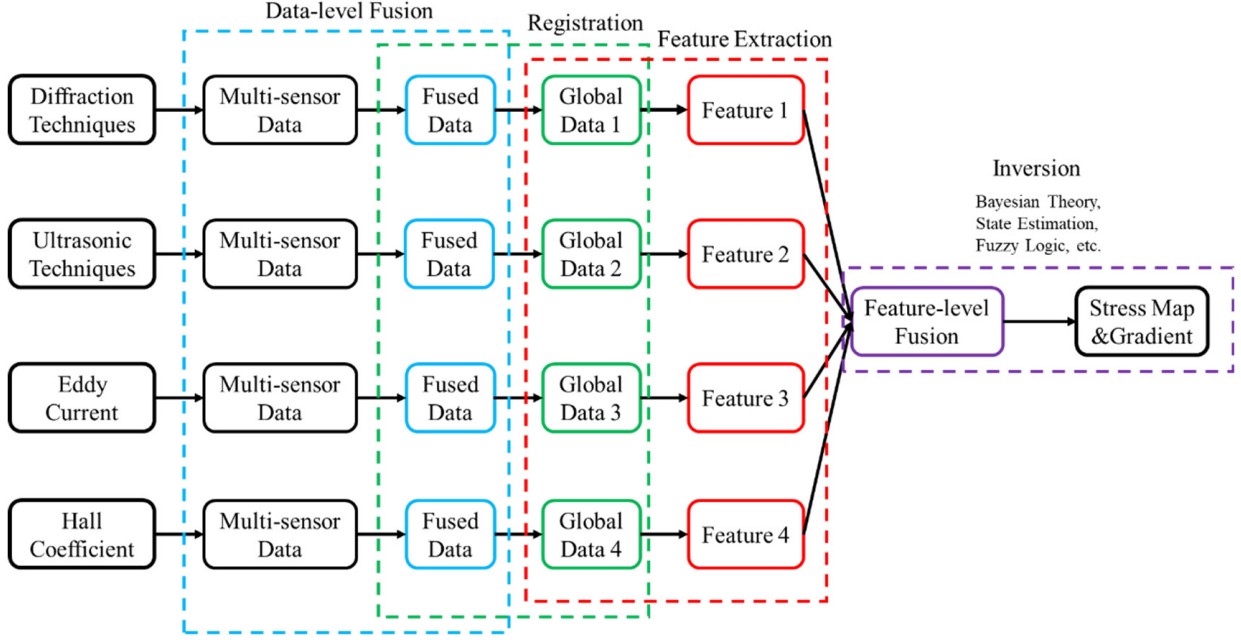

**Figure 12.** Flow chart of NDE data fusion.

Upon registration, these data are subjected to feature extraction to obtain multiple features sensitive to residual stress, which will be of help in building a comprehensive model of residual stress. Next, feature-level fusion of these features needs to be done to obtain the exact model for stress measurement. In the inversion process, some classical data fusion models can be applied, such as Bayesian theory, state estimation, and fuzzy logic, and the method selection will be dependent on the actual situations/cases. In addition to fusion strategy represented in Figure 12, the fusion data can also be processed independently after data-level fusion to obtain decisions for different NDE techniques. These decisions can then be fused in the decision level to produce the ultimate states of residual stresses.

Although there are some attempts to measure residual stress using data fusion [182–184], it is still a challenging problem to collect useful information from data and make robust decisions. There are competing effects other than residual stress content in the signal, and the noise may vary. Additionally, potential conflicts between the data from different NDE techniques may exist, and the data may be incomplete and lack simplicity. These are all relevant issues in the nature of the collected data [185,186]. Nevertheless, comprehensive residual stress profiling (including stress map over test area and stress gradient along

penetration depth) which can be attained upon data fusion techniques are well applied, considering the specific physics and optimal fusion scheme.

### 6.2. Strengthened Stress Prediction Models Based on Machine Learning

Various mechanical properties of metallic components, including microhardness, roughness, and residual stress, have been proven to be predictable based on artificial neural network (ANN) [187–192]. Figure 13 displays the structure of such a prediction model, which mainly consists of an input layer, hidden layers, and an output layer. Factors that may affect mechanical properties, such as processing parameters, assumptions, and constraints, etc., can be used as inputs to the model, and the corresponding parameter $P_i$ is selected as the input layer neuron. The prediction of nonlinearity is accomplished by different neurons $Y_j$ in the hidden layer and the weights $\omega_{ij}$. The output layer converts the hidden layer activation into the desired output $R_k$. Based on this, a training set containing the influencing parameter $P_i$ and the mechanical property parameter $R_k$ can be constructed experimentally. The ANN model allows for learning the mapping relation between inputs and outputs via iterations, making it capable of predicting the targeted mechanical properties.

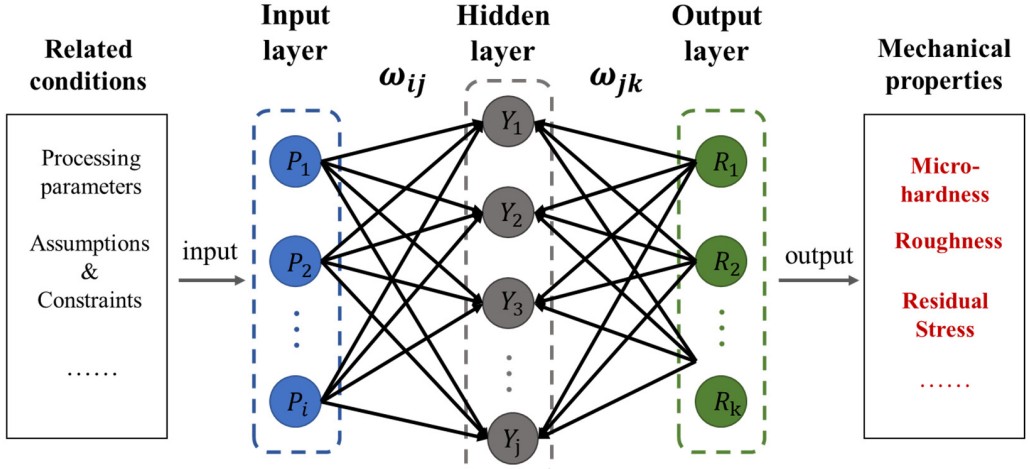

**Figure 13.** The structure of mechanical properties prediction model based on the artificial neural network.

Specifically, ANN-based models can provide enhanced assessment of the residual stresses. For instance, such methods have been applied to the shot peening process in order to predict the effect of associated main processing parameters, i.e., Almen strength and surface coverage, on the treated steel samples' deformation layer depth, surface residual stresses, and residual stress maximum [193]. Through the trained models, the residual stress maximum and surface mean were well reconstructed. A similar ANN prediction technique has been exploited to characterise the mechanical properties of an example LSP-treated TC4 titanium alloy and further investigate the residual stress distribution patterns [194]. In addition, a generative adversarial network (GAN) model has been developed, which obtained physical fields, such as stress or strain, directly from the material microgeometry map with high accuracy [65,195].

Besides ANN-based models, more recently, a so-called "Sim-to-Real" approach featuring transfer learning has been proposed to strengthen the experimental database from simulations by eliminating bias between the simulated data and the experimental data. The concept of "Sim-to-Real" aims to transfer knowledge learned from simulations to the real world to address the simulation-to-reality-gap. Such an approach is used to improve the target learner's performance on the target domain, i.e., experimental data by transferring knowledge contained in a different but related source domain, i.e., simulated data. Figure 14 schematically illustrates the process of scatterer wavefield prediction by subtracting the baseline obtained from numerical simulations via "Sim-to-Real" technique in guided wave-based damage detection. A series of studies have been conducted to

reconstruct the experimental baselines from the simulated guided wave fields. Hopefully, the "Sim-to-Real" can conquer the insufficient experimental data problem and improve the accuracy of data-driven prediction for mechanical properties, including residual stresses.

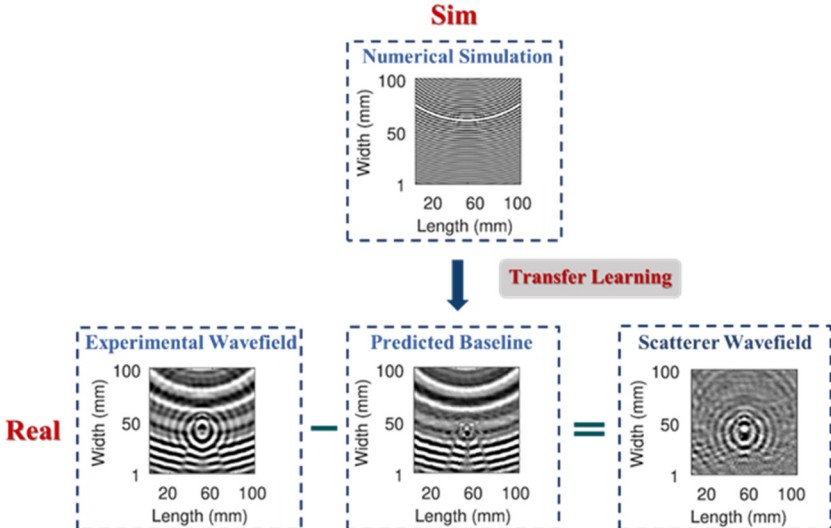

**Figure 14.** Wavefield prediction based on the "Sim-to-Real" approach [196].

*6.3. Potential SHM Strategy for Residual Stresses*

In industrial NDE, routine inspections are often carried out at plant shutdowns, using removable transducers and instrumentations, and the measurements are infrequent and performed under different testing environments [197]. Structural health monitoring (SHM), by contrast, instruments permanent attached transducers to enable frequent measurements during the operation. The monitored data have the comparability to the previous accessed data due to the consistency of the testing environment. Referring to SHM philosophy, the states of residual stresses of critical regions in metallic components can also be monitored long-term. Although there seems to be no direct SHM systems for residual stresses reported in the literature, extensive research efforts have manifested the SHM applications. SHM techniques have been investigated in damage detection. Croxford et al. (2007) proposed a feasible ultrasonic guided wave SHM strategy for defects detection with temperature compensation [198]. Worden et al. (2007) raised several basic and universal axioms for SHM techniques and addressed that the assessment of damage requires a comparison between two system states [199]. As per the SHM implementation routines, baseline subtraction method and intelligent feature extraction techniques are currently available to interpret the frequently acquired SHM data, which show the consistency of the axioms and sustains the feasibility of the potential SHM strategy for residual stress profiling. However, two immediate hindrances need to be solved—reliable permanently-attached sensing and management of large data flows generated by frequent interrogation.

Active sensing is essential to SHM technology for residual stress. Sensors excite signals actively to monitor the structure deterioration conditions. Restricted by harsh working environment in aircraft engines, including high temperature and high pressure, regular sensors cannot sustain. Among all the NDE techniques reviewed earlier, ultrasonics-based and eddy current approaches show the most promise for working in harsh environments. Permanently installed ultrasonic systems for crack monitoring were proposed by Kande (2010) [200] and Cegla et al. (2011) [201], and the ultrasonic sensors are capable of working at relatively high temperatures (e.g., 550 °C). The eddy current sensors and corresponding monitoring methodology were also investigated [202]. In addition to the impacts on active sensors, harsh operating environment directly affect signals by changing temperature or pressure, as well as their interpretation. The "Sim-to-Real" technique mentioned ear-

lier may be beneficial in building a relatively accurate and reliable baseline and in data interpretation [196].

Another main hindrance of previous routine SHM research is the lack of attention to handle the large data flows [203]. Since SHM data would be generated much more frequent than normal inspections, the ability to interpret terabytes of data is required. The data need to be re-organized and cleaned before analysing [197,204]. Due to the rapid advancement of deep learning and more extensive immediate research efforts in Smart NDE, the data could be analysed by a well-trained prediction model for improving data utilization and interpretation accuracy.

## 7. Discussions

The most widely used non-destructive technique for residual stress measurement is diffraction. The limited penetration depth of conventional X-ray makes this technique destructive in order to obtain deeper residual stress information because of layer removal of materials. Although the synchrotron X-ray diffraction technique has satisfying resolution and penetration depth, there are limited synchrotron X-ray sources globally, which restricts the routine residual stress measurement due to the high cost and the long lead time, just as the Neutron diffraction technique, which has comparable penetration capability and compromised resolution. The limitations of the diffraction techniques stimulate the development of other non-destructive residual stress profiling methods, such as Ultrasonic, eddy current, and Hall coefficient, that exploit the stress-dependence of certain material properties. The ultrasonic technique is most explored due to extensive research and understanding of this area. However, the rest of the techniques can still find their unique applications. Table 1 lists the comparisons of the reviewed non-destructive residual stress profiling techniques. The spatial resolution and penetration depth of the reviewed techniques are illustrated in Figure 15.

**Table 1.** Comparisons of the non-destructive residual stress profiling techniques.

| Technique | Material Type | Portability | Advantages | Limitations |
|---|---|---|---|---|
| X-ray Diffraction | Crystalline | No | Small gauge volume<br>Bi-axial measurements<br>Widely available | Limited penetration depth<br>Accuracy seriously affected by<br>grain size and texture<br>Semi-destructive for bulk measurement<br>Surface preparation required |
| Synchrotron X-ray Diffraction | Crystalline | No | Good penetration depths<br>Tri-axial residual stress measurements<br>Small gauge volume (typically < 1 mm$^3$)<br>Applicable to complex shapes<br>Indifferent to surface finish | Elongated gauge volume<br>Only applicable to polycrystalline materials<br>Accuracy affected by grain size and texture<br>Very long lead time |
| Neutron Diffraction | Crystalline | No | Good penetration depths<br>Tri-axial residual stress measurements<br>Applicable to complex shapes<br>Indifferent to surface finish | Only applicable to polycrystalline materials<br>Accuracy affected by grain size and texture<br>Very long lead time<br>Not suitable for surface measurements |
| Critically refracted longitudinal wave | Solid | Yes | Quick measurement<br>Greatest sensitivity to residual stress<br>Frequency-dependent penetration depth | Dramatically influence by microstructure |
| Rayleigh wave | Solid | Yes | Quick measurement<br>Frequency-dependent penetration depth | Dramatically influence by microstructure |
| Eddy current | Conductor | Yes | Quick measurement<br>Frequency-dependent penetration depth | Selectivity to residual stress |
| Hall coefficient | Conductor | Yes | Quick measurement<br>Frequency-dependent penetration depth | Selectivity to residual stress |

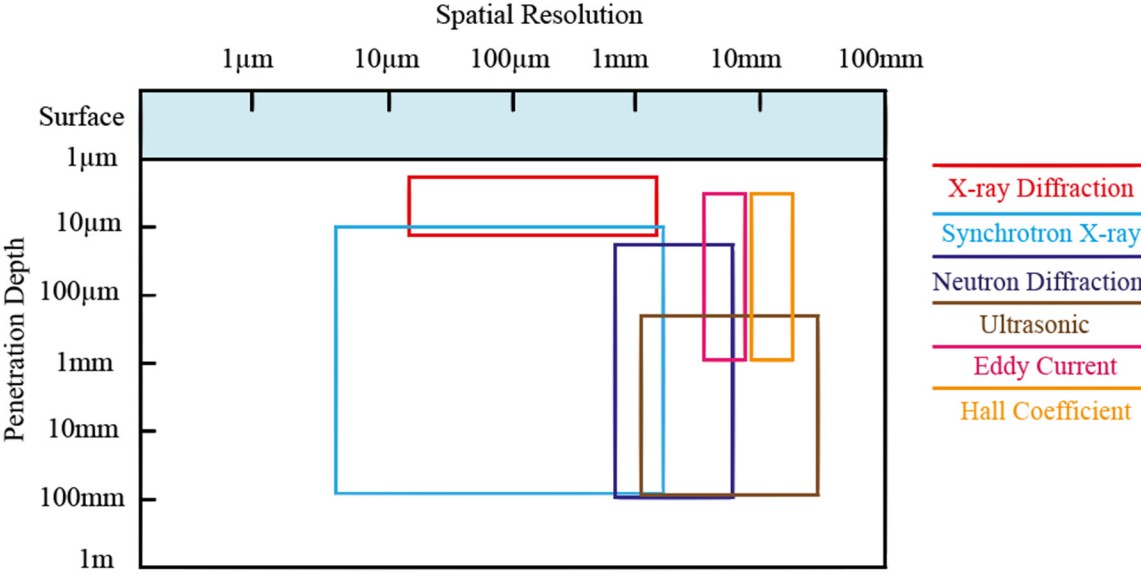

**Figure 15.** Penetration depth and spatial resolution of reviewed techniques.

## 8. Conclusions

The main conclusions of this review can be summed up as follows:

1. The diffraction techniques are now mature and well-established for residual stress measurement with widespread applications and usually act as a calibration tool to validate the results obtained by other techniques. Synchrotron and neutron experiments can be performed in either reflection or transmission configuration with a monochromatic or white beam (full spectrum) source [65,205]. Synchrotron X-ray diffraction and neutron diffraction methods are capable of 3D mapping of residual stresses thanks to their perfect penetration depth. However, the very high cost and limited access to the facilities restrict these two methods as being only applicable to a few components of interest.

2. Critically refracted longitudinal wave, Raleigh wave, eddy current, and Hall coefficient all exhibit frequency-dependent penetration depth and, therefore, are suitable for depth-dependent residual stress measurement. A common challenge for these techniques is the competing effects other than residual stress.

3. Ultrasonic testing methods using $L_{CR}$ or Rayleigh waves are sensitive to residual stress. Ultrasonic methods can measure the average stress volumetrically and obtain the residual stress at varied depths through inversion schemes. It is worth noting that crystallographic texture, surface roughness, and cold work will also influence the ultrasonic velocity measurements, which remains a challenge for accurately characterising residual stresses via ultrasonics-based techniques.

4. Eddy current technique has been demonstrated to be successful for residual stress profiling in certain nickel-based superalloys, such as IN100 and Waspaloy, in the past two decades. However, it has also been found that in other aircraft engine materials such as IN718 and Ti64, the residual stress influence on the conductivity is obscured or even overshadowed by cold work.

5. The Hall coefficient technique has been proposed recently, and it is anticipated that it can be applied to more materials than eddy current because the cold work influence on Hall coefficient is in the same direction as residual stress in the recently studied aircraft engine materials. The high-frequency inductive sensing for Hall coefficient measurement has been validated very recently and can potentially be exploited for sub-surface residual stress profiling. The dual-mode Hall impedance and eddy current conductivity measurements enable the feasibility of separating the competing residual

stress and cold work contributions in surface-enhanced aero-engine materials. The Hall coefficient technique has great potential, yet still requires more research efforts.

6. The separation of residual stress information requires a quantitative understanding of the coupling factors such as cold work, surface roughness, microstructure, etc. Due to the different sensitivities of these competing factors by the reviewed techniques, it is worth establishing inspection protocols that employ a combination of non-destructive techniques to obtain a more accurate and reliable residual stress profile.

7. The selectivity rather than sensitivity is a more important consideration for non-destructive residual stress profiling techniques. An exclusively stress-related parameter will be extremely valuable for the development of a new non-destructive technique for residual stress profiling of aero-engine components.

8. Three categories of future research trends are proposed, including data fusion of different NDE technique, strengthened prediction models based on machine learning, and potential SHM strategy, aiming at improving the accuracy, efficiency, and reliability of residual stress profiling. Nevertheless, more dedicated research efforts are required.

**Funding:** This work was funded by the National Natural Science Foundation of China [grant number 12004026], National Key Research and Development Project (Grant No. 2018YFA0703300), the Young Elite Scientist Sponsorship Program by China Association for Science and Technology [grant number 2020QNRC002], and the Fundamental Research Funds for the Central Universities [grant number YWF-22-L-1054].

**Conflicts of Interest:** The authors declare no conflict of interest.

## Nomenclature

| | |
|---|---|
| $d$ | Lattice spacing of the material under X-ray diffraction test |
| $\theta$ | Diffraction angle |
| $\lambda$ | X-rays' wavelength |
| $\varepsilon$ | Elastic strain |
| $\Delta d$ | Deviation of lattice spacing under residual stress |
| $\Delta\theta$ | Deviation of diffraction angle |
| $\psi$ | Azimuth angle of the measurement system |
| $d_0$ | Stress-free lattice spacing |
| $h$ | Planck constant |
| $c$ | Speed of light in vacuum |
| $E_p$ | Energy of the photon |
| $p$ | Neutron momentum |
| $m_n$ | Neutron mass |
| $v_n$ | Travelling speed of the neutron |
| $T$ | The time of flight of the neutron |
| $L_n$ | Flight path of the neutron |
| $\rho$ | Density |
| $\lambda_0$ and $\mu_0$ | Lamé constants |
| $m_0$, $l_0$ and $n_0$ | Murnaghan constants |
| $\varepsilon_i$ | Elastic Strain Components. |
| $v_{ii}$ and $v_{ij}$ | Velocity of ultrasonic wave |
| $\Delta\sigma$ | Stress variation |
| $E$ | Young's modulus |
| $L$ | Acoustoelastic constant |
| $t_0$ | Travel time at the stress-free state |
| $\Delta t_T$ | Travel time change due to the temperature change |
| $\sigma_{ij}$ | Averaged stress in each layer |
| $\sigma_i$ | Varied stresses |
| $D_i$ | Varied penetration depths of the incident $L_{CR}$ wave |
| $K_{12}^1$, $K_{12}^2$, $K_{21}^1$ and $K_{21}^2$ | Acoustoelastic coefficients with the superscripts denote the loading directions |
| $v_{SAW}$ | Acoustic velocity of SAWs |

| | |
|---|---|
| $f_{SAW}$ | Generated SAW frequencies |
| $\lambda_{gs}$ | Grating space |
| $\beta$ | Nonlinearity parameter of Rayleigh wave |
| $f_{rayleigh}$ | Driving fundamental frequency of Rayleigh wave |
| $\delta$ | Penetration depth of eddy current |
| $f_c$ | Frequency of eddy current |
| $\mu_c$ | Permeability of eddy current |
| $\sigma_c$ | Conductivity of eddy current |
| $\tau_c$ | Isotropic plane stress |
| $K_{\parallel}$ and $K_{\perp}$ | Parallel and normal electro-elastic coefficients |
| $\gamma$, $\gamma'$ and $\gamma''$ | Phase and precipitates of nickel-based superalloys |
| $R_H$ | Hall coefficient |
| $V_H$ | Hall voltage |
| $I$ | Current passing through the sample |
| $B$ | Normal magnetic flux density |
| $t$ | Sample thickness |
| $n_c$ | Carrier density |
| $q$ | Carrier type |
| $P_i$ | Input layer neuron |
| $Y_j$ | Different neurons |
| $\omega_{ij}$ | Weights used in the ANN |
| $R_k$ | Output |

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
