# Peer review of "A Review of Non-Destructive Evaluation (NDE) Techniques for Residual Stress Profiling of Metallic Components in Aircraft Engines"

_aerospace, doi:10.3390/aerospace9100534_

Round 1

Reviewer 1 Report

The authors have made a very good job reviewing this important topic: it is easy to read the paper, including recent advances and technical and fundamental challenges associated to particular approaches, and they propose a logical and believable outlook for the field. I support its publishing. Comments:

1. I suggest to add one additional section about the characteristics of residual stress: how does it develop through the thickness, what are typical values, how does it develop in different materials, what accuracy we need to describe it. 

2. There is the Barkhausen noise technique mentioned in the introduction but I did not find it in the main text.

3. Paragraph starting from line 777 needs more explanation, what is "Sim-to-real". 

4. Can you mention any working residual stress SHM applications. You need to measure the same spot with multiple techniques to get reliable data. Is it achievable?

Typos:

Line 662 "The and plastic ..."

Reviewer 2 Report

The proposed work is really perfect. Especially, the residual stress-based review on aircraft engines is really a trendsetter. A few of the suggestions are given below that will further enhance the quality of this article. 

1. In the abstract, the authors are mentioned the below statement:

"The insufficient penetrating capability of the only currently available non-destructive residual stress assessment technique X-ray diffraction has prompted an active search for alternative non-destructive techniques."

My suggestion is to add furthermore explanations under the introduction section, which will support your aforesaid statement. Especially, the lag of this work has failed to provide enough support for your shortlisted NDT technique. Please concentrate on this issue. 

2. Almost authors incorporated 16 equations, in which more than 30 parameters are included. therefore, I request the authors to develop a new table for nomenclatures. 

3. Few of the references are in a different language than English. Is it acceptable for this focused journal? please check and do the needful accordingly. 

4. Commonly review-based papers have the main issues, which have been arrived from copyrights. Please ensure about your 14 figures. Are they developed by current authors or extracted from the references? if yes, please put the reference number over it. Also please get the proper permission from the sources. 

5. What is the need to fix Table 1 and Figure 14 under the conclusion? Both table 1 and Figure 14 are well constructed so please incorporate them in the discussion section (before the conclusions). This will attract the proposed work in a great manner. 

6. Question marks should be avoided in the headings of Table 1. 

7. Since the reference section plays a major role in the review paper, the referred articles are cited properly in the main and other sections. Especially, the format to represent the reference in the main text is: "first author name et al. (YEAR)" and the format to represent the references under the REFERENCE section is: 1. "first three author names with et al" or 2. "please type complete authors list". As per my suggestion, please check your article. 

8. unused abbreviations can be avoided. I found this error in this article. Please check and do the needful. 

9.  I found another mistake that the same symbols and/or notations are used for different equations. Please clarify it. 

Round 2

Reviewer 2 Report

All the needful corrections are made. Thanks for your support and actions.